# Designing the Composition of Cement Stabilized Rammed Earth Using Artificial Neural Networks

**DOI:** 10.3390/ma12091396

**Published:** 2019-04-29

**Authors:** Hubert Anysz, Piotr Narloch

**Affiliations:** Faculty of Civil Engineering, Warsaw University of Technology, Al. Armii Ludowej 16, 00-637 Warsaw, Poland

**Keywords:** rammed earth, cement stabilized rammed earth, artificial neural networks, sustainable building material

## Abstract

Cement stabilized rammed earth (CRSE) is a sustainable, low energy consuming construction technique which utilizes inorganic soil, usually taken directly from the construction site, with a small addition of Portland cement as a building material. This technology is gaining popularity in various regions of the world, however, there are no uniform standards for designing the composition of the CSRE mixture. The main goal of this article is to propose a complete algorithm for designing CSRE with the use of subsoil obtained from the construction site. The article’s authors propose the use of artificial neural networks (ANN) to determine the proper proportions of soil, cement, and water in a CSRE mixture that provides sufficient compressive strength. The secondary purpose of the paper (supporting the main goal) is to prove that artificial neural networks are suitable for designing CSRE mixtures. For this purpose, compressive strength was tested on several hundred CSRE samples, with different particle sizes, cement content and water additions. The input database was large enough to enable the artificial neural network to produce predictions of high accuracy. The developed algorithm allows us to determine, using relatively simple soil tests, the composition of the mixture ensuring compressive strength at a level that allows the use of this material in construction.

## 1. Introduction

Technologies for erecting buildings using raw earth were popular in all civilizations of the world until the beginning of the 20th century. Nowadays, every third person in the world lives in a house built from the earth while in the developing countries the number is more than half of the population [1]. Thanks to the widespread availability of earth, numerous objects made from it are located on every inhabited continent (Figure 1).

Striving for the most extensive use of natural materials, buildings made of earth can be an alternative option to other natural materials such as wood, which due to its flammability must be impregnated, whereas earth is non-flammable [3]. On the other hand, building structures made of earth are considered to not be durable. However, it is worth noting that in many regions of the world impressive ancient buildings made of earth can be admired. The best-known example is the Great Wall of China (Figure 2), of which numerous fragments erected in the rammed earth technology about 4000 years ago have survived until the present day.

In the regions of the world with hot and dry climates, there are not only buildings, but whole cities built with technologies using the earth. The most well-known city of this type is Shibam in Yemen (Figure 3). Some buildings of this city, erected in the 19th and 20th centuries, with technologies using the earth, are even 11 stories tall. This confirms that it is possible to use the land as a material for building multi-story residential buildings. Other examples of constructions built with technologies using the earth that belong to the history of architecture, including the proportions of soil, cement, and water that were used in the past, can be found in references [4,5].

Over the centuries, many building technologies that use earth have been developed in various regions of the world. Their variety results, among other factors, from the type of land and climate conditions prevailing in the area [6]. Construction techniques using earth can be divided into three main groups [7]:
-Techniques that use earth in load-bearing monolithic constructions.-Techniques that use earth in load-bearing masonry structures.-Techniques that use earth as a non-bearing construction material in combination with a supporting structure of another material.

This paper focuses on one of the technologies of erecting load-bearing monolithic earth structures i.e., cement stabilized rammed earth (CSRE). CSRE is a highly sustainable construction technique characterized by low energy demand and a small amount of waste generated during the construction process. The structure consists of rammed layers of moist mixture made in a formwork set on a stable foundation (Figure 4). A basic component of the mixture is the subsoil that lies under the layer of humus usually taken directly from the construction site. The limiting factor determining the suitability of the subsoil for rammed earth technology is the quantity of organic substances. The organic soil is biodegradable, easily absorbs water, and is highly compressible. Therefore, its presence in the soil mixture should not exceed 1% of the total weight of the soil [8]. Depending on the particle size distribution of the soil, the mixture is modified by adding appropriate aggregate fractions and Portland cement. The components are mixed together in an air-dry state, and then water is added to ensure the proper moisture content of the mixture. Layers of the mixture are filled in the formwork and then compacted by mechanical means. After proper compaction of the layer, successive ones are added until the planned height of the element is reached. The formwork is then removed. Since compaction leads to closer packing of soil grains, this process eventually leads to increased mechanical strength [4]. Durability of a monolithic wall made of CSRE with adequate load capacity depends on the particle size of the soil used and the addition of Portland cement [9]. With CSRE technology it is possible to construct monolithic walls that contain communication openings, since erecting walls in layers provides the possibility of easy assembly of reinforcement [10].

A significant part of building project expenses is associated with transport of materials to the construction site. Therefore, one of the ways to reduce those costs is to reduce the need for transportation. Cement stabilized rammed earth technology allows for erecting monolithic load-bearing walls using locally available material, which is a very good solution in less urbanized areas located far from wholesalers.

The compressive strength is a significant mechanical feature of a structural building material. The results of laboratory tests for the compressive strength of cement stabilized rammed earth are presented in numerous publications (Table 1). However, results obtained by individual authors are difficult to compare due to differences in:
-Shapes of the tested samples,-Energy used in ramming samples and the related volume density of samples,-Particle size distribution of the earth used in the mixture,-Mineral and chemical composition of the earth,-Moisture content of the earth mixture,-Moisture content of the samples at the time of the test,-Content and type of Portland cement,-Duration and conditions of samples storage before strength tests.

The lack of information on some of those parameters is the main challenge in analyzing the results published. For example, from the tests included in Table 1, only article [11] lists the chemical composition of the earth. In references [12,13] it is presented that differences in the chemical composition of the clay used can have a significant impact on the obtained value of compressive strength.

Determining the proper moisture content of the soil-cement mixture is one of the key issues in rammed earth technology. The moisture content determines its workability and, as a result, affects the most important properties of the material, including its compressive strength. The optimum moisture content (OMC) for a rammed earth mixture is critical in achieving maximum dry density through dynamic compaction [19]. If too little water is in the mixture, then the soil cannot achieve the same level of compaction due to the greater degree of friction between the soil particles [20]. If too much water is added, then capillary water occupies the soil pore spaces reducing the level of achievable compaction and increasing the level of porosity in ready dried wall [20]. The optimum moisture content of a CSRE mixture depends on the particle size of the soil [19,21], as well as the cement content and compaction method used [22]. The data presented in the Table 1 shows that for a similar method of sample compaction, the optimum moisture content increases along with the amount of the cement added. The density of the samples made of CSRE mixtures with a similar moisture and cement content depends primarily on the soil particle size.

Although it is possible to determine precisely the optimum moisture content of the mixture under conditions found during construction, the problem is that these conditions are highly variable for longer periods of time. They are influenced by, among other factors, atmospheric conditions, that are usually unpredictable. It is worth noting that this phenomenon has been noticed in the standards. Currently, the largest accumulation of knowledge in modern rammed earth construction is thought to exist within Australia and New Zealand [19]. The New Zealand Standard [23] allows for moisture content to differ by 3% from the planned optimum. According to the authors of the article, the range of a soil mixture’s moisture content allowed by this standard is too wide. This can be confirmed by the test results [24], where the compressive strength values of samples from soil mixtures with a moisture content of 2% lower or 2% higher than the optimal value differ from each other by 50%. As a consequence of the high tolerance in the moisture content of the CSRE mixture, the compressive strength to be used in the design of a standard earth wall construction is only 0.5 MPa [25], regardless of the compressive strength results of the CSRE samples obtained in laboratory tests. It is worth noting that according to reference [23], for a series with a minimum of 5 samples with a ratio of height to width of 1:1, the lowest value of compressive strength should be greater than 1,3 MPa, and therefore more than 260% of the design compressive strength value.

Designing concrete is itself a complex process. However the methods of design and factors critical for the compressive strength are well known. Applying CSRE as a structural material is, for the time being, an experimental process. The amount of cement is much lower in CSRE than in concrete and therefore a difference of, e.g., 5 kg of cement for 1 m^3^, influences the properties of CSRE much more than the properties of concrete. In general, CSRE seems to be a construction material that is very sensitive to changes in composition, where the aggregate skeleton and the way of compaction play very important in reaching the required compressive strength. The correlation factors for the compressive strengths of samples calculated for water content and cement to be added show that these variables are weakly or not at all correlated with compressive strength. Even the Spearman’s correlation factor calculated for the w/c ratio and compressive strength (for 373 CSRE samples described further in this paper) is not significant (−0.622).

As the absolute value of the correlation factor with a single component of CSRE (water or cement) is below 0.5, its influence on the compressive strength is meaningless. The absolute value of the correlation of the water-to-cement ratio is higher than 0.5, so the influence of compressive strength can be observed, but is still not a strong one. The higher w/c is, the lower the compressive strength. These Spearman’s correlation values confirm that the other properties of CSRE mixtures also have a significant influence on the compressive strength.

A possible way of finding a composition of a CSRE mixture that meets compressive strength criterion is a prediction based on the obtained results of the samples’ tests. In concrete, the type and amount of cement used are of crucial importance to the compressive strength obtained. In the case of CSRE, the amount of cement is smaller, so that the remaining properties of the mixture, such as particle size distribution, humidity, energy, and the method of ramming, as well as the mineral composition of the applied earth significantly influence the obtained compressive strength. In the absence of standards for the design of cement stabilized rammed earth and the individual approach of researchers to this problem, the authors propose the use of artificial neural networks (ANN) to determine the desired composition of a mixture. The more the nature of the process is unknown and the greater the complexity of the process, the more benefits using ANN provides (compared to statistical models, expert systems, Figure 5) [26,27].

According to the aforementioned figure, Artificial Neural Networks (ANN) are widely applied as a predictive tool in the construction industry. They have been applied for predicting:
-Delays in the completion date of construction works [28,29],-Construction costs [30,31],-Level of deterioration of multi-story buildings [32],-Loss of a client in the case of collusion [33],-Energy demand for housing purposes [34,35],-Financial results of construction companies [36].

Properties of construction materials can be predicted with the use of ANN [37,38]. There are successful examples of ANN use in designing specific concrete where recycled aggregates were also considered [39]. The above mentioned application of ANN is also a solid base for expecting good results in the CSRE mixture designing.

## 2. Materials and Methods 

### 2.1. Database of CSRE Test Results

#### 2.1.1. Materials

Hall and Djerbib [19] point out that for sample production, if soils obtained from different locations were to be used, it would be difficult to analyze the laboratory test results due to the large number of variables introduced, including but not limited to the mineralogical composition. Therefore, in laboratory tests inorganic soil mixture of clay, sand, and gravel was obtained from the construction site. Each of these components were then dried to a constant mass and mixed in ten proportions to obtain particle-size distribution curves shown in the Figure 6. Each soil was named numerically in relation to its sand: gravel: silty clay ratio out of a total 10 components. For example, symbol 613 means that the mixture consists of 6 parts sand, 1 part gravel, and 3 parts clay by weight. For each of these mixtures, 3 to 10% by weight of Portland cement CEM I 42.5R was added. Afterwards, water was added to these mixtures to ensure an optimum moisture content, i.e., the moisture of the mixture, which, guaranteed the highest dry density for the adopted method of ramming the samples. For comparative purposes, the tests were also carried out on mixtures with a moisture content 2% higher and 2% lower than the optimal level.

#### 2.1.2. Preparation of Samples

CSRE samples were prepared in cubic molds, 100 mm × 100 mm × 100 mm. The formation of the samples was carried out by ramming the mixture into three equal layers using a 6.5 kg rammer. The parameters of the rammer were selected on the basis of New Zealand Standard [23]. The samples were formed by freely lowering the rammer from a height of 30 cm to the surface of the moist mixture. For this method of compaction, the optimum moisture content in individual mixtures was 7% to 10% depending on the particle size distribution and the amount of cement.

For each series, 10 samples were made. The authors participated in the preparation of all samples. Therefore, the influence of the sample preparation method on the compressive strength results can be ruled out. The samples to be tested were demolded after 24 h. Then they were cured for 27 days in a condition of relatively high humidity of 95% (±2%) and temperature of 20 °C (±1 °C).

#### 2.1.3. Results of Testing the Samples

Since the rammed earth keeps the layered structure, the samples were loaded in the direction of the ramming. 373 compressive strength results were obtained for the prepared samples. The results of some of these samples are shown in Table 2. The results obtained served as a database for calculations using ANN.

The cement content in the samples of CSRE varied from 3% to 10% and their moisture content from 6% to 14%. The mean compressive strength was 6.00 MPa with a standard deviation of 2.093 MPa. The compressive strength varied from 2.400 to 13.011 MPa (Figure 7).

All tested samples were analyzed and tested. Their features are summarized in Table 3. Appendix A (Table A1) comprises the full set of data.

Previous analysis [28,40,41] was the basis of the decision to standardize the input data with the linear method. The following Formula (1) was used:(1)$$a_{i}=\frac{a_{0i}}{maxa_{0i}}\;for\;1\leq i\leq 373$$
where: $a_{i}$—the standardized value of a (from the i experiments); $a_{0i}$—the original value of a feature observed for i experiment.

The standardization process–done separately for each feature–completed the preparation of data for feeding them to ANN.

### 2.2. Proposed Solution

#### 2.2.1. Algorithm

The authors propose an algorithmic procedure consisting of five steps that make it possible to design the composition of the CSRE. In laboratory tests, it is possible to determine the moisture of the CSRE mixture by adding an appropriate amount of water to the mixed dry ingredients of the mixture. Practically, CSRE requires adaptation of the mix design method to the real conditions prevailing at the construction site. The unique parameters of a construction site include its soil particle-size distribution and the moisture content in the soil.

The authors propose the algorithm (Figure 8) in which the laboratory test results can be used to determine the composition of the soil mixture through the use of field tests. From these tests, the particle-size distribution of the soil (content of clay, silt, sand and, gravel fractions) and its moisture content are obtained (STEP I). In the process of preparation of the mixture on the construction site, it should be considered that the addition of water depends on the natural moisture of the soil. In order to design CSRE mixture with a certain compressive strength based on the subsoil used, it is necessary to determine:
-The amount of Portland cement to be added,-The amount of water,-The ramming energy needed to obtain a specific dry density of the compacted soil mixture.

The following features are assumed in the proposed calculation algorithm (STEP II):
-The amount of cement to be added,-The density of compacted sample,-The compressive strength.

The density of the compacted mixture is a function of the energy used for compaction. The study of this obvious relation is not the subject of this article, however, such a relationship was observed during the experiments. Therefore the density of the compacted sample is used as one input of the ANN and it represents the energy used for the compaction process. Next (STEP III), the moisture content of the mixture intended for compaction is calculated using the artificial neural network (ANN). The inputs to the ANN are: the particle-side distribution (percentage content of clay, silt, sand, and gravel fractions), and the values of the three aforementioned assumed parameters. In STEP IV, the addition of water is calculated based on the determined amount of cement, the assumptions, and the local moisture of the soil. In the next step (STEP V), the method requires the preparation of a CSRE sample with a designed composition and checking if it can be compacted to the assumed dry density. If so, the recipe for preparing the CSRE element with the assumed compressive strength is determined. If the assumed density cannot be achieved, then the calculation of the moisture content should be repeated (back to STEP II), taking the density obtained in STEP V. The procedure between STEP II and STEP V should be repeated until the assumed density is achieved. If it cannot be obtained as a result of the procedure repetition, the assumed amount of cement to be added to the mixture should be modified. Then, the artificial neural network will predict a different moisture content.

#### 2.2.2. Creating the Artificial Neural Networks

A multi-layer perceptron (MLP) ANN consists of neurons (nodes)—see Figure 9—where input values are transformed by a so-called activation function giving the value as an output from the neuron.

The argument of the activation function is calculated as shown in Formula (2). Inputs are multiplied by weights and summed up.
(2)$$f\left(\overset{n}{\underset{i}{\sum }}x_{i}*w_{i}\right)=wy$$
$x_{i}$—the value of input i to the neuron; $w_{i}$—the weight assigned to input i to the neuron; wy—the value of activation function i.e., output from the neuron.

The most commonly applied activation function is a logistic Formula (3).
(3)$$f\left(x\right)=\frac{1}{1+e^{-x}}$$
e—the base of natural logarithm.

But other functions are applied too e.g., hyperbolic tangent, linear function.

In MLP, ANN neurons are grouped in layers: an input layer, a hidden layer (one or more), and an output layer (see Figure 10).

The activation function is not applied in the nodes of the input layer. Observing a given phenomenon, the data (parameters accompanying the phenomenon) is collected, as well as the results. When N sets of data are collected, together with N results, the ANN can be fed with them. The metaheuristic algorithms (built in the software) try to find the set of weights (described above) that minimize the error i.e., the difference between real effect of the phenomenon and the output calculated by ANN for all N cases. The accuracy of predictions achieved from ANN can be verified by comparing them to the observed real cases. The most often applied type of error used for evaluating the accuracy of predictions is the mean squared error (MSE) which is calculated from the Formula (4):(4)$$MSE=\frac{\sum_{i=1}^{N}{\left(c_{i}-r_{i}\right)}^{2}}{N}$$
$c_{i}$—predicted value calculated for i sample; $r_{i}$—real value observed during i observation; N—number of observations.

To do so, a portion of cases from the collected dataset should be excluded for the verification ANN accuracy (so called validation). Another portion from the dataset that should be excluded is the part called the test dataset. It provides protection from overtraining (overfitting) ANN. The remaining part of collected data is called a training dataset, as it is used for finding the weights. While the built-in algorithm searches for the best weights, the MSE calculated for training dataset is continuously compared to the MSE for the testing dataset. When the MSE for the testing dataset starts to rise, it defines the end of the training process [42]. If the increasing continues, it could lead to overfitting, as shown in Figure 11.

Then the set of weights found is kept and ANN is ready to use for predictions. The accuracy is evaluated based on the MSE for validating the dataset, which is not used in the training process at all.

## 3. Results

As an output from the ANN, the moisture content was chosen as the dependent variable. The other seven types of data checked for each sample (i.e., clay content, silt content, sand content, gravel content, cement content, density and compressive strength) serves as input–independent variables. The ANN predictions were calculated with the use of STATISTICA software (ver. 13, Dell Inc., Round Rock, TX, USA). The data were randomly divided into three subsets (training, testing, validating) with a proportion of 70:15:15 [43], so in the validating subsets there were 55 samples. The software allows the application of only one hidden layer. The features of five of the best predicting ANN (MLP type) found are presented in Table 4.

The small differences between correlation factors for training, testing and validating datasets (and original values of the moisture content) ensure that the results found are not overfitted. The differences between the original and projected values of the moisture content are shown in Figure 12 (results for all 5 nets and all subsets).

Before the prediction errors were calculated, the prediction values (for the validating dataset) had been back converted to have the original unit i.e., moisture content given in a percentage. The MSE and the square root of it, called the RMSE (root mean squared error), are good for comparing the accuracy achieved by different ANNs. Nevertheless, the three other types of errors can be more informative for a researcher who would like to use predicting features of ANN for a new set of data where the result is unknown. These types of errors are as follows (5,6,7):
-Mean absolute error (MAE)
(5)$$MAE=\frac{\sum_{i=1}^{N}\left|c_{i}-r_{i}\right|}{N}$$
-Mean absolute percentage error (MAPE)
(6)$$MAPE=\frac{\sum_{i=1}^{N}\left|\frac{c_{i}-r_{i}}{r_{i}}\right|}{N}*100\%$$-Maximum absolute percentage error (maxAPE)
(7)$$maxAPE=\left|\frac{c_{i}-r_{i}}{r_{i}}\right|*100\%\;for\;1\leq i\leq N$$

The values for the above-mentioned errors are shown in Table 5 for all five of the best predicting ANNs found.

The lowest value of error is highlighted for each error type. The ANN No. 2 (7-6-1) was chosen for further predictions as three types of errors calculated (for the validating dataset) are the lowest. The net No. 5 gives the lowest maxAPE. We decided to omit it, based on the fact that the accuracy of predictions made for the lowest and the highest moisture content (as can be seen in Figure 12) are the poorest. Observing the best MSE, MAE and MAPE for ANN No 2, it can be said that extreme moisture content barely influenced the training process. Based on the chosen ANN (7-6-1), the predictions of desired moisture content can be obtained.

## 4. Discussion

Below is the exemplary process for designing CSRE based on the proposed algorithm and ANN predictions.

STEP I. Let us assume (for the purpose of exemplary calculation) that according to the Figure 8, the soil found at the location of a future structure made of CSRE is characterized by the following aggregate fractions content: clay 10.5%, silt 20.1%, sand 49.4%, gravel 20.0%. The original moisture content checked was 3.00 [%].

STEP II. Real values (from STEP I) are to be extended by assuming the following properties: the compressive strength to achieve (e.g., 7.885 MPa), the addition of cement (e.g., 9%), and the density after the ramming process (e.g., 2238 kg/m^3^).

STEP III. The chosen ANN (7-6-1) produced the required moisture content at 7.89 [%]. Then, considering the original moisture content of the soil (3%), the addition of the water can be calculated.

STEP IV. Considering the original moisture content of the soil from the construction site (3.00 [%]), the mass of water to be added to obtain the desired moisture content of the CSRE mixture with the assumed cement addition, can be calculated based on the Formula (8):(8)$$m_{x}=\frac{y}{1-y}\left(m_{g}+m_{c}\right)-m_{w}$$
where: $m_{g}+m_{c}$—the mass of moist subsoil from the construction site, [kg]; $m_{g}$—the mass of the dried subsoil, [kg]; $m_{w}$—the mass of water in the subsoil [kg] determined as the difference in mass of the moist subsoil and the dried subsoil, [kg]; $m_{c}$—the mass of the cement added, [kg]; $m_{x}$—mass of water that should be added to the CSRE mixture to achieve the desired moisture content specified by the ANN, [kg]; y—predicted moisture content of the CSRE mixture with the use of ANN, [%].

For 100 [kg] of subsoil with a moisture content of 3%, the mass of dry soil and the mass of water in the soil can be calculated from the following Formula (9):(9)$$w=\frac{m_{w}}{m_{g}+m_{w}}$$

Thus, the mass of water in this soil is:(10)$$3\%=\frac{m_{w}}{100}\;\to \;m_{w}=3\;\left[\mathrm{kg}\right]$$

Then, the mass of dry soil can be calculated:(11)$$m_{g}+m_{w}=100\;\to \;m_{g}=97\;\left[\mathrm{kg}\right]$$

For this dry soil, the mass of cement to be added is: (12)$$m_{c}=97\times 9\%=8.73\;\left[\mathrm{kg}\right]$$

To obtain the moisture content of the CSRE mixture containing 100 [kg] of subsoil with a moisture content of 3% and 8.73 [kg] of cement added, the following amount of water should be added:(13)$$m_{x}=\frac{7.89\%}{1-7.89\%}\left(97+8.73\right)-3≅6.06\;\left[\mathrm{kg}\right]$$

STEP V. As the process of ramming the mixture influences the density of the compacted CSRE, and it is not yet standardized, predictions were made for the set of assumed density, varying from 2118 to 2358 [kg/m^3^]. The predicted moisture content results are shown in Figure 13.

The above density-moisture curve has been found by predicting the moisture content for a given density through the use of the ANN. The density can be checked after adding all the components of the CSRE and compacting the mixture. If the pair of parameters (density and moisture) fits any point on the curve presented in Figure 13, the assumed compressive strength should be achieved. The algorithm then comes to a stop. The composition of the CSRE based on the site’s subsoil is found. If any combination of the predicted moisture content and the assumed density is not possible to achieve (the pair moisture-density does not fit any point on the curve), the assumptions should be changed–the algorithm comes back to STEP II.

The proposed method requires that, the following simultaneous changes to the assumptions of three values should be made (in STEP II): required compressive strength, cement addition, and the density to achieve. These changes may cause some trouble for the user. The density may be evaluated before the assumption is made based on trial soil compaction where only water is added. The compressive strength and the cement added is positively (but not strongly) corelated, as stated above. The following Table 6—based on the 373 samples prepared and checked—shows the ranges of compressive strength achieved for different portions of cement added.

It is suggested to base the three assumptions on the 373 experiments done by the authors and summarized in Table 6, as well as on the subsoil on site (checking the density after compaction with only water added).

## 5. Conclusions

The article presents the possibility of using artificial neural networks for designing cement stabilized rammed earth mixtures. Using artificial neural networks, the moisture content of the CSRE mixture was predicted to ensure the assumed compressive strength. The input database was large enough that the artificial neural network found by the software Statistica produced predictions of high accuracy. The mean absolute error MAE was 0.565 [%] of the designed moisture content. It should be emphasized that the prediction errors obtained are smaller than the accuracy when preparing the CSRE mixture under the conditions prevailing at the construction site (e.g., due to changing weather conditions). The algorithm was created for designing cement stabilized rammed earth with the use of subsoil obtained from the construction site. The developed algorithm allows us to determine, using relatively simple soil tests (analysis of grain size and humidity), the composition of the mixture ensuring compressive strength at a level that allows the use of this material in construction. It should be emphasized that the equipment necessary to assess density and particle size distribution is more simple, accessible, and affordable to provide on site, than equipment for compressive strength testing. Software for artificial neural networks (not commonly used in finding a CSRE recipe) that excludes the need of using compressive strength test machinery, is a step toward the real application of CSRE as a structure material. The variety of the soil types that can be found, and—for the time being—the low popularity of CSRE, create some uncertainty in the compressive strength level that can be achieved on a building site. The large size of the database of the samples tested, as well as high accuracy of ANN predictions achieved and presented above, made the authors believe that the created method of designing CSRE will be successfully applied for random soils gathered from variety of locations. The forthcoming research is to be aimed at simplification of the process of designing the CSRE mixture in a way that would hopefully allow builders to prepare the recipe on site.

## Figures and Tables

**Figure 1 materials-12-01396-f001:**
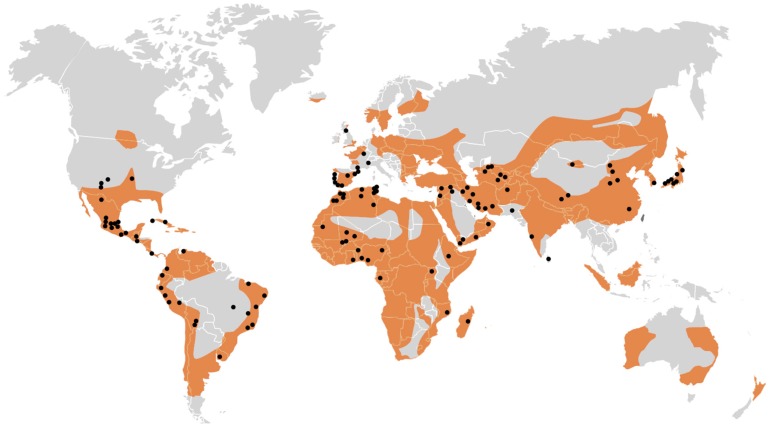
Areas of occurrence of numerous buildings made of earth (yellow color). The black points marked the earth architecture inscribed in the UNESCO World Heritage List [2].

**Figure 2 materials-12-01396-f002:**
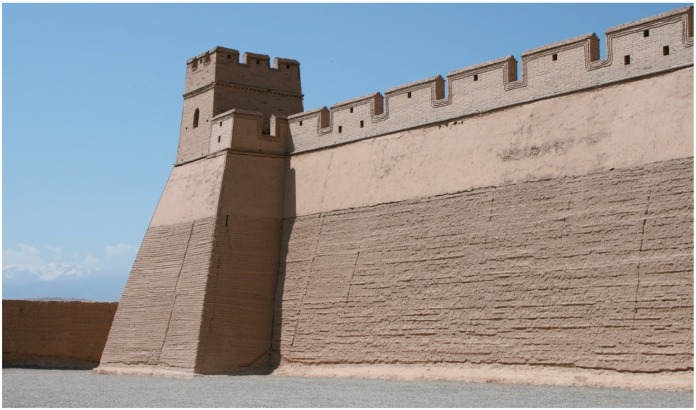
The Great Wall of China erected with rammed earth technology.

**Figure 3 materials-12-01396-f003:**
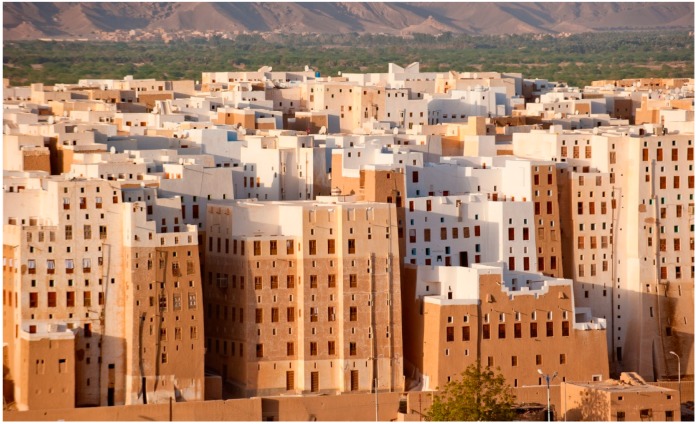
City of Shibam, Yemen. The city whose buildings were mostly erected with construction technologies using earth.

**Figure 4 materials-12-01396-f004:**
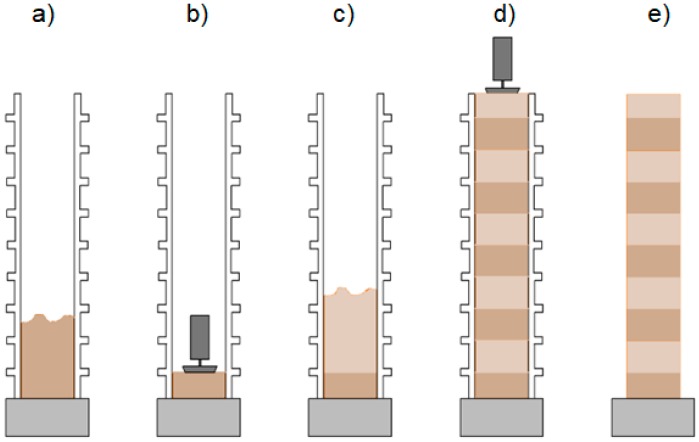
A scheme for erecting a wall using rammed earth technology. Different layers of the CARE monolithic wall are marked with different colors. Steps when erecting a wall: (**a**) Formwork is built and filled with a layer of moist soil-cement mixture. (**b**) The layer of moist mixture is compressed. (**c**) The next layer of moist soil-cement mixture is added. (**d**) Successive layers of moist earth are added and compressed. (**e**) The formwork is removed leaving the monolithic CSRE wall.

**Figure 5 materials-12-01396-f005:**
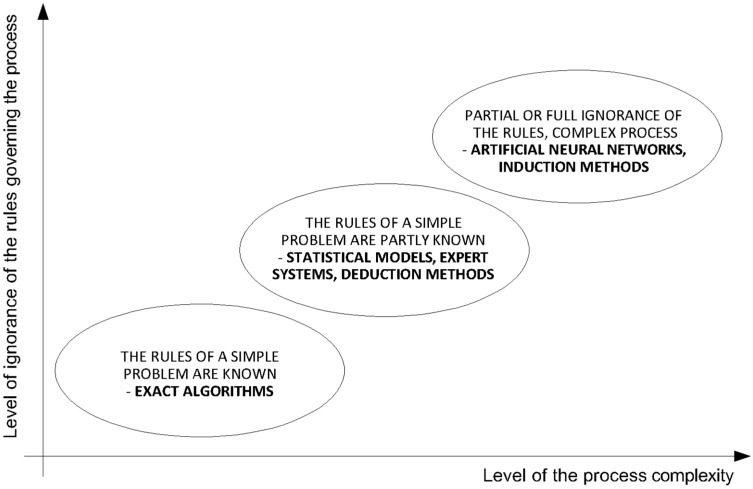
Features of the problems where applying ANN brings more accurate results in solving the problem [26,27].

**Figure 6 materials-12-01396-f006:**
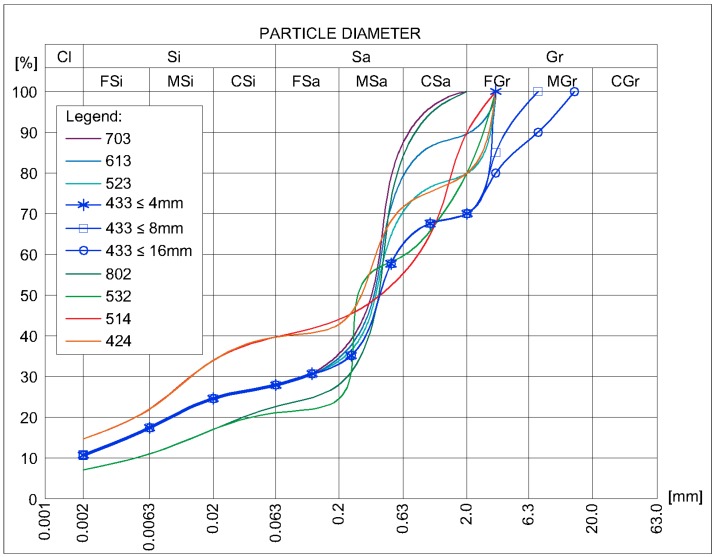
Particle size distribution of soils used for laboratory tests.

**Figure 7 materials-12-01396-f007:**
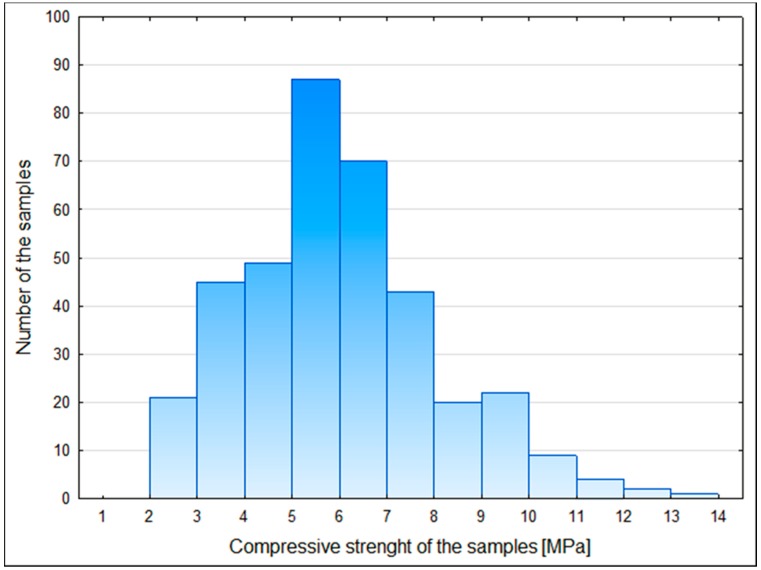
Histogram of the compressive strength of the samples.

**Figure 8 materials-12-01396-f008:**
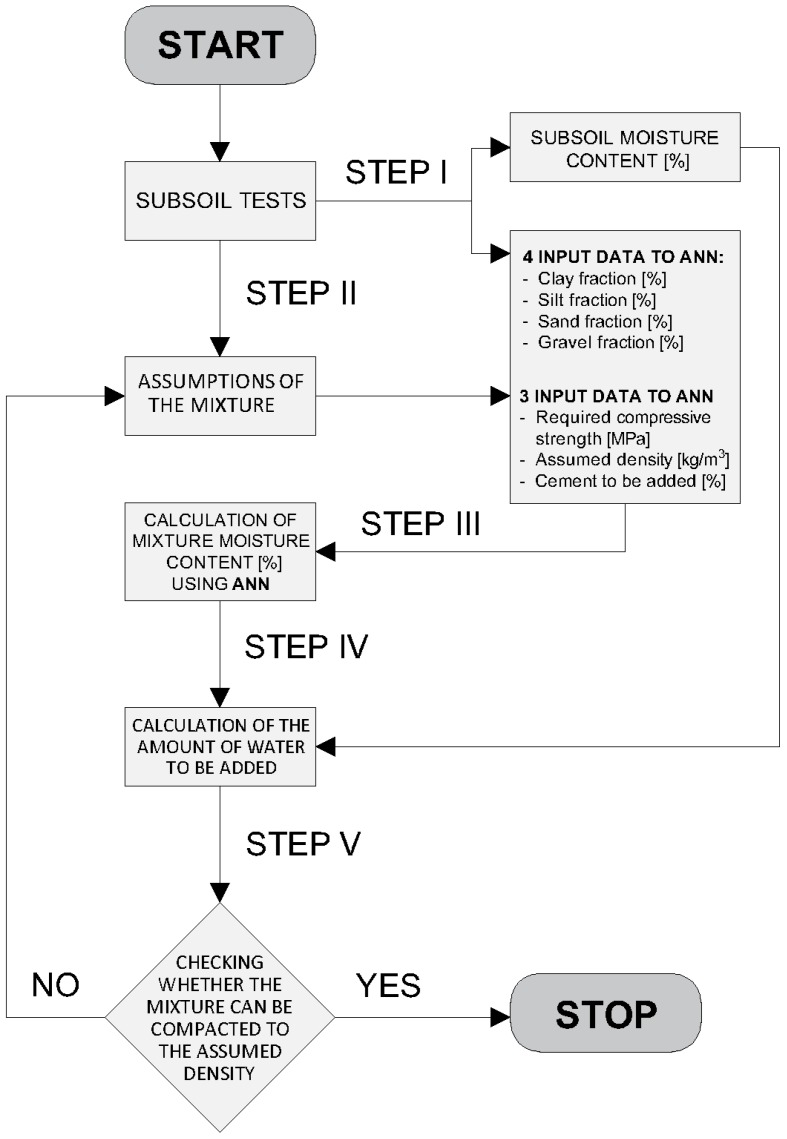
The algorithm for determining the composition of the soil mixture based on field tests.

**Figure 9 materials-12-01396-f009:**
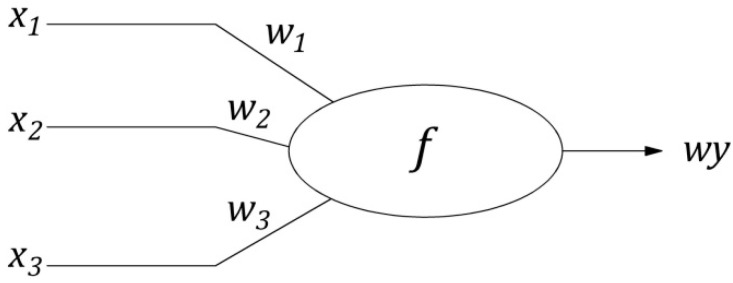
The scheme of a neuron (node) with three inputs and one output where $x_{1},x_{2},x_{3}$ are input signals from the previous ANN’s layer, $w_{1},w_{2},w_{3}$ are their weights, wy the value of activation function i.e., outgoing signal from the node to consecutive layer or output of the net [28].

**Figure 10 materials-12-01396-f010:**
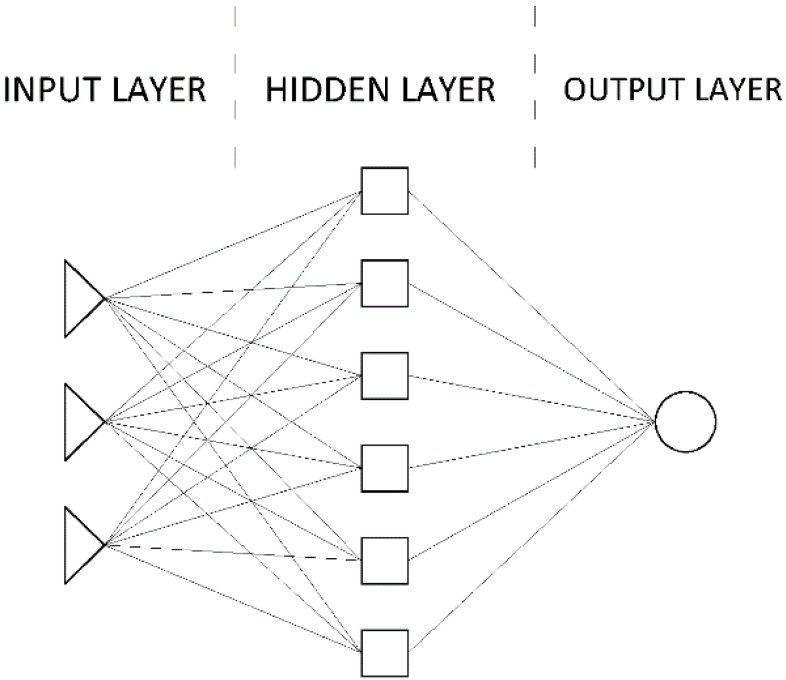
Exemplary ANN (multi-layer perceptron, MLP) with 3 input nodes, 6 nodes in one hidden layer, 1 output node. The ANN of this architecture is marked as (3-6-1).

**Figure 11 materials-12-01396-f011:**
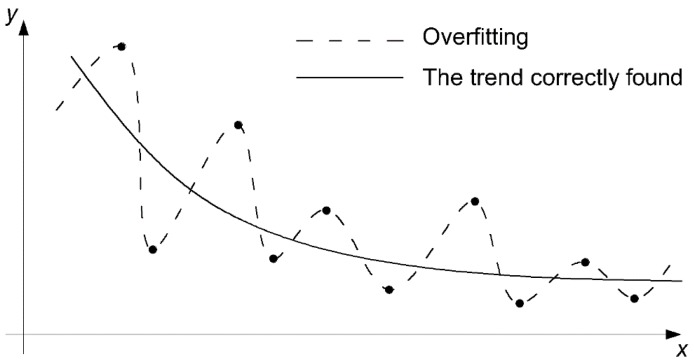
Overfitting predictions and the trend correctly found [28].

**Figure 12 materials-12-01396-f012:**
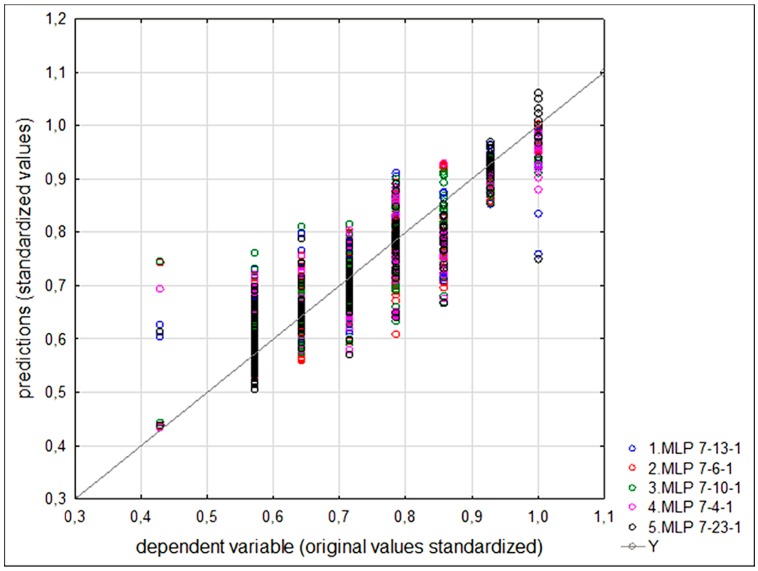
Original and predicted values of the moisture content (both standardized).

**Figure 13 materials-12-01396-f013:**
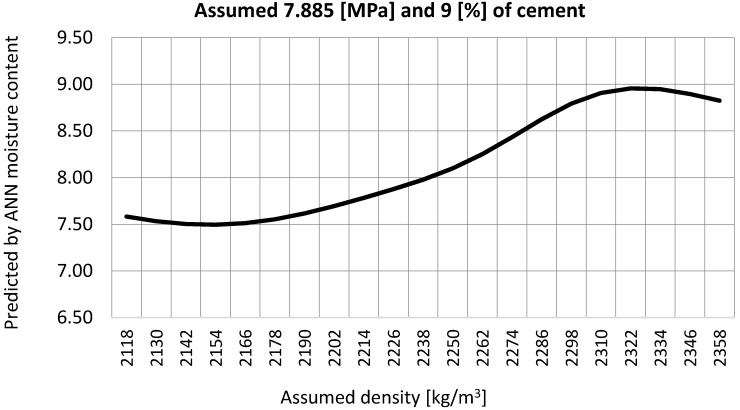
Predicted final moisture content for given density varied from 2118 to 2358 [kg/m^3^] for required compressive strength 7.885 [MPa] and assumed 9 [%] addition of cement.

**Table 1 materials-12-01396-t001:** Compressive strength results of CSRE from various references [11,14,15,16,17,18].

Reference	Particle Size Distribution of Earth	Shape and Dimensions of Samples [cm]	Moisture Content of the Mixture [%]	Compaction Method	Density of Samples [kg/m^3^]	Cement Addition	Curing Conditions	Compressive Strength [MPa]
[11]	Sand 64% Silt 18% Clay 18%	cuboid 10 × 10 × 20	9.5	hydraulic press (15 MPa)	Max. 1877	8%	27 days, 70% RH	18.4 (AVG.)
[14]	Gravel < 19 mm Gravel 32% Sand 66% Clay + Silt 2%	Cylinder d = 10, h = 20	7.8	pneumatic rammer	-	8%	2 days in forms, 7 days tightly wrapped, 19 days in an air-dry condition	10 (AVG.)
			8.2					10–12 (AVG.)
			8.5					7–8 (AVG.)
[15]	Gravel 26–50% Sand 46–70% Silt + Clay 4 %	Cylinder d = 10/15, h = 20/30	8.85–9.15 (OMC)	pneumatic rammer	2091	10%	28 days in an air-dry condition, 13–25 °C	13.8 (AVG.)
[16]	Gravel 45% Sand 40% Silt 5% Clay 10%	wall element 50 × 50 × 11	5.4 (OMC)	pneumatic rammer	2156–2139	4.5%	7 days tightly wrapped, min. 20 days in an air-dry condition	6.68 (AVG.)
[17]	>19 mm 4.3% Gravel 32.2% Sand 59.4% Clay +Silt 8.4%	wall element 100 × 65 × 16	9.5–11.0 (OMC)	Mechanized rammer, layers 15 [cm] high	1800–2000	6% 8% 10%	In an air-dry condition	2.06 (CH) 2.94 (CH) 3.09 (CH)
	>19mm 17.9% Gravel 56% Sand 29.6% Silt + Clay 14.4%					6% 8% 10%		1.69 (CH) 1.64 (CH) 2.35 (CH)
	>19mm 6.4% Gravel 50.5% P 30.4% π + I 19.1%					6% 8% 10%		1.52 (CH) 1.72 (CH) 1.92 (CH)
[18]	Gravel 20% Sand 50% Silt 25% Clay 5%	cylinder d = 15 h = 30	5.3 (OMC)	pneumatic rammer	2005–2012	4.5%	6 days in an air-dry condition	5.25 (AVG.)
							12 days in an air-dry condition	16 (AVG.)

AVG.—average value of compressive strength; CH—characteristic value of compressive strength; OMC—optimum moisture content; RH—relative humidity.

**Table 2 materials-12-01396-t002:** Results of compressive strength for some chosen samples.

Sample Number	Particle Size Distribution [%]				Cement Addition [%]	Moisture Content [%]	Density of the Sample [kg/m^3^]	Compressive Strength [MPa]
	Clay ϕ > 0.002 [mm]	Silt ϕ ≥ 0.002 ϕ < 0.063 [mm]	Sand ϕ ≥ 0.063 ϕ < 2.0 [mm]	Gravel ϕ ≥ 2.0 ϕ < 4.0 [mm]				
6	0.105	0.192	0.403	0.300	3	9	2295	4.01
81	0.105	0.192	0.403	0.300	6	8	2240	6.45
108	0.105	0.210	0.585	0.100	6	8	2218	5.43
125	0.105	0.210	0.585	0.100	6	10	2299	5.55
144	0.105	0.219	0.676	0.000	6	10	2265	5.16
179	0.140	0.244	0.416	0.200	9	10	2301	5.97
219	0.105	0.192	0.403	0.300	9	9	2274	5.91
245	0.105	0.192	0.403	0.300	9	12	2240	5.55
328	0.070	0.176	0.754	0.000	9	8	2241	9.75
364	0.105	0.192	0.403	0.300	10	13	2282	7.06

**Table 3 materials-12-01396-t003:** The summary of the features’ values calculated for all 373 samples.

Feature	Minimum	Maximum	Mean Average	Standard Deviation
Clay [% of the total weight of aggregates]	0.070	0.140	0.107	0.015
Silt [% of the total weight of aggregates]	0.149	0.253	0.206	0.021
Sand [% of the total weight of aggregates]	0.403	0.754	0.516	0.109
Gravel [% of the total weight of aggregates]	0.000	0.300	0.172	0.117
Cement addition [% of the total weight]	3.00	10.00	7.57	2.11
Moisture content [% of the total weight]	6.00	14.00	9.86	1.74
Density [kg/m^3^]	2054.00	2406.33	2250.33	56.43
Compressive strength [MPa]	2.40	13.01	6.00	2.09

**Table 4 materials-12-01396-t004:** The best predicting ANN found.

ANN Number	No. of Neurons in the Hidden Layer	Correlation of the Training Dataset	Correlation of the Testing Dataset	Correlation of the Validating Dataset	Activation Function in the Hidden Layer	Activation Function in the Output Layer
1	13	0.9104	0.8877	0.8557	tanh	linear
2	6	0.9031	0.8798	0.8831	tanh	linear
3	10	0.9233	0.8817	0.8522	tanh	tanh
4	4	0.9053	0.8626	0.8503	tanh	linear
5	23	0.9431	0.9183	0.8579	tanh	exponential

**Table 5 materials-12-01396-t005:** The errors of predictions calculated.

ANN No.	1	2	3	4	5
MSE	0.996	0.796	0.951	0.976	0.928
MAE	0.744	0.565	0.657	0.722	0.668
MAPE [%]	8.19	6.52	7.34	8.03	7.16
maxAPE [%]	46.03	73.28	73.75	61.79	43.38

**Table 6 materials-12-01396-t006:** Results of compressive test of 373 samples of CSRE.

Cement Addition [%]	Number of Samples	Compressive Strength [MPa]			
		Min	Max	Mean av.	Standard dev.
3	36	2.438	4.200	3.215	0.456
6	120	2.400	8.741	5.439	1.304
9	174	2.649	13.011	6.874	2.232
10	43	4.030	9.450	6.357	1.557

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
