# Peer review of "Designing the Composition of Cement Stabilized Rammed Earth Using Artificial Neural Networks"

_materials, 2019, doi:10.3390/ma12091396_

Reviewer 1 Report

In my opinion, this paper is based on a very complete experimental work. Moreover, the proposed application of artificial neural networks for in-field design of composition of rammed earth seems very useful to reduce time, efforts and risks associated to this eco-efficient construction system. Nevertheless, I believe that the paper structure is not adequate, it is not easy to read and it does not show the value of the experimental work behind.

The paper results section is centred on the ANN results, but lacks from an experimental validation of its usefulness in real applications and also should take into account the effect of changes in mineralogical composition of the soils in these real applications with respect to those used for the ANN database. So, my advice would be to centre the paper on the ANN set up and not on its use or real application, taking advantage of all the information collected in the database and its importance for usefulness of the ANN.

Taking this opinion into account, I have several comments to improve the paper.

Regarding the introduction, I have minor comments:

-          To avoid misunderstandings, it should be indicated in the text that Table 1 contains references from 2 to 9 before citing reference 10 in line 65.

-          In the first paragraph of page 4, the argument to use standards from New Zealand as a reference should be stated.

-          Table 2 is not necessary, as the information contained in this table may be easily included in the text. In addition, a comment on how these correlation data have been obtained would be useful.

Regarding the rest of the paper, may comments imply major changes, as I think that the structure is not correct. In fact, I would suggest considering the data for the database as results and present a complete description of these data to support the argument of being representative and useful as database for ANN set up.

Considering this point of view, I would suggest the following modifications of the Materials and methods section:

-          The section should start with ANN description (current 2.2.2 sub-section). In addition, I suggest changing the first paragraph, from lines 199 to 210, to the paper introduction and, in my opinion figures 5, 7 and 9 are not necessary. I think that it is enough if proper general references are cited for readers needing more details in ANN.

-          From the previous description of ANN, it is clear that database is necessary, so description and preparation conditions of the database may be included in a second sub-section, but avoiding including any result in this sub-section.

-          Finally, the specific algorithm used may be included as in current 2.2.1 sub-section.

Consequently, my proposal for the results section would be:

-          Start with a complete description of the database results. The results from each set of conditions (particle size distribution, cement content and moisture content) should be included in current table 3 with mean value and standard deviation of density and compressive strength. The discussion of these results should assess that the database is representative and useful for ANN generation.

-          Selection of the best ANN would be the following sub-section.

-          After this, the current section 4 would be included as an application example.

-          If possible, a case study could be added, with experimental data to validate the ANN use in real applications.

Conclusions should be modified according to the new perspective and further validation of the ANN usefulness could be proposed as future work, together with the analysis of the effect of chemical composition of the soils.

Author Response

The authors are grateful to the Reviewer for appreciation of their effort, as well as, for valuable comments and suggestions.

Comment 1:  In my opinion, this paper is based on a very complete experimental work. Moreover, the proposed application of artificial neural networks for in-field design of composition of rammed earth seems very useful to reduce time, efforts and risks associated to this eco-efficient construction system. Nevertheless, I believe that the paper structure is not adequate, it is not easy to read and it does not show the value of the experimental work behind.

Response: Now, our effort is emphasized also in the abstract (primarily it is done in the Appendix). The structure of the paper is given as the template, but specific issues related to the structure (stated in the comments below) are applied.

Comment 2:  The paper results section is centred on the ANN results, but lacks from an experimental validation of its usefulness in real applications and also should take into account the effect of changes in mineralogical composition of the soils in these real applications with respect to those used for the ANN database. So, my advice would be to centre the paper on the ANN set up and not on its use or real application, taking advantage of all the information collected in the database and its importance for usefulness of the ANN.

Response: The authors realize that the influence of mineralogical composition can influence the compressive strength. The most influential are clay minerals. The samples, in laboratory conditions, were prepared with the same type of clay. The content of minerals is modelled partially by the different content of clay fraction (among 373 samples). Nevertheless we cannot conclude about mineralogy issues based on the tests presented in the manuscript.

Comment 3:  To avoid misunderstandings, it should be indicated in the text that Table 1 contains references from 2 to 9 before citing reference 10 in line 65.

Response: In the revised version, references appear in correct order.

Comment 4:  In the first paragraph of page 4, the argument to use standards from New Zealand as a reference should be stated.

Response: In the revised version, the argument to use standards from New Zealand as a reference is  stated.

Comment 5  Table 2 is not necessary, as the information contained in this table may be easily included in the text. In addition, a comment on how these correlation data have been obtained would be useful.

Response: Previous Table 2 is removed. Their content is described in text. The information that correlation is calculated based on our 373 tests is added. We believe that there is no need to present the formula for Spearman’s correlation factor.

Comment 6: Regarding the rest of the paper, may comments imply major changes, as I think that the structure is not correct. In fact, I would suggest considering the data for the database as results and present a complete description of these data to support the argument of being representative and useful as database for ANN set up.

Considering this point of view, I would suggest the following modifications of the Materials and methods section:

-          The section should start with ANN description (current 2.2.2 sub-section).

Response: As the template force us to apply “Materials and Methods” chapter, it is natural to describe:

The database created

The algorithm invented, where data are processed

ANN basics, as it is important part of the algorithm invented.

Let us remain the aforementioned structure unchanged.

In addition, I suggest changing the first paragraph, from lines 199 to 210, to the paper introduction

Response: This section (originally in lines 199 to 210) is shifted to the introduction now and it is lightly modified.

(…) and, in my opinion figures 5, 7 and 9 are not necessary. I think that it is enough if proper general references are cited for readers needing more details in ANN.

Response: Figure, previously marked as 7, is removed. Let us remain figures (previously marked as) 5 and 9. They provide the reader more familiar with the reason of ANN application and ANN idea itself. Based on our experience, scientists familiar with CSRE are not necessarily familiar – at the same level – with ANN tool. We try to make the paper easy understandable for a wider range of readers (even if advanced tool is used – ANN) and the drawings on subject support this aim.

-          From the previous description of ANN, it is clear that database is necessary, so description and preparation conditions of the database may be included in a second sub-section, but avoiding including any result in this sub-section.

Response: The content of (previous) point 3.1 is shifted to the chapter 2 where data are described (to avoid describing data in results).

-          Finally, the specific algorithm used may be included as in current 2.2.1 sub-section.

Response: The changes done accordingly to Reviewer’s kinds remarks, made us remaining the algorithm in subsection 2.2.1 (as Reviewer suggested).

Comment 7: Consequently, my proposal for the results section would be:

-          Start with a complete description of the database results. The results from each set of conditions (particle size distribution, cement content and moisture content) should be included in current table 3 with mean value and standard deviation of density and compressive strength. The discussion of these results should assess that the database is representative and useful for ANN generation.

Response: Now, all issues concerning the data are grouped in subsection 2.1 “Database …”. The description of data is extended giving the idea (without the need of detailed checks in appendix) what are the values of independent variables and dependent one.

 -          Selection of the best ANN would be the following sub-section.

-          After this, the current section 4 would be included as an application example.

-          If possible, a case study could be added, with experimental data to validate the ANN use

Response: Let us leave the „Results” chapter as it is proposed now. The result – 5 best ANN found by a software, based on assumed by the authors conditions – needed to be discussed. The one, the best ANN had to be chosen, so we proved our choice in this chapter. As try to be stick to the template of the paper, the application example we place as “Discussion”. It is to emphasize that the sub-set of validating data comprise the real data (features of samples). These samples were prepared in laboratory conditions, but they are real. So, the high accuracy of predictions (in proposed method) is proved. Validation of the proposed method on random soil samples, gathered from different regions of our country is a separate issue, that we plan as a next hundreds of test, next analysis. We realize that this way of confirming the proposed method – confirmed in laboratory condition so far – is desirable. We state it in the summary.

Comment 8: Conclusions should be modified according to the new perspective and further validation of the ANN usefulness could be proposed as future work, together with the analysis of the effect of chemical composition of the soils.

Response: As a result of rewriting the paper where substantial part of your advises is considered, we modified the conclusions to match the structure and content of the paper in the revised form.

Reviewer 2 Report

Summary

This article is a study demonstrating the potential to use artificial neural networks (ANN) to design cement stabilized rammed earth (CSRE) compositions. With multiple inputs and highly varying outputs, this is a reasonable method to use to make these types of predictions, and thus I believe the topic of this article is worth publishing about. This is the core of this article, and I am hardly an expert in machine learning techniques, so I have little to comment on much of this article, but I do have some comments that could be of use. The method seems a bit unwieldy, requiring a large input database for training and validation (the authors produced 373 mixtures) but these databases could be obtained in the future as more data gets online with respect to construction materials and processes. This has high potential.

Review

As stated above, I am hardly an expert in machine learning. My understanding of this method is that you could take variable inputs (soil conditions at the site) and produce a specific output (in this case, a CSRE mixture). The example the authors show is to perform a calculation for required water addition to a specific soil with moisture content, and fixed cement addition. It would seem to me that one could also vary two parameters, say both moisture added and cement addition. Is this a possibility? From a practical standpoint, that’s what I would want - just to be able to define 1) how much water and 2) how much cement I would need to achieve a particular density and compressive strength (especially strength, I would imagine).

How do you account for ramming energy? I would think that this is something that would be highly variable across different job sites. Is there any systematic study that has been done that would allow you to incorporate this also in the ANN?

Finally, as a minor item, - in the manuscript, the equations are shifted across two or more lines and it makes it difficult to read, please fix this. 

Author Response

The authors are grateful to the Reviewer for appreciation of their effort, as well as, for valuable comments and suggestions.

Comment 1:  This article is a study demonstrating the potential to use artificial neural networks (ANN) to design cement stabilized rammed earth (CSRE) compositions. With multiple inputs and highly varying outputs, this is a reasonable method to use to make these types of predictions, and thus I believe the topic of this article is worth publishing about. The method seems a bit unwieldy, requiring a large input database for training and validation (the authors produced 373 mixtures) but these databases could be obtained in the future as more data gets online with respect to construction materials and processes. This has high potential.

Response: Authors are grateful to the Reviewer for his/her positive and encouraging comments.

Comment 2: My understanding of this method is that you could take variable inputs (soil conditions at the site) and produce a specific output (in this case, a CSRE mixture). The example the authors show is to perform a calculation for required water addition to a specific soil with moisture content, and fixed cement addition. It would seem to me that one could also vary two parameters, say both moisture added and cement addition. Is this a possibility?

Response: Based on our experience in applying ANN tool, we avoid predicting more than one category of an output (e.g. water and cement content). The error of prediction (for the validating set of data) is a sum of squared errors divided by the number of samples in validating set (MSE). If there are two categories of output it is impossible to discuss what part of the error comes from water content and what comes from cement content predicted. MSE then, can be useful for comparing the prediction accuracy between the nets found, but not for evaluating preciseness of amount of water to be added or cement to be added. That is the reason of our choice to predict only the final content of water. Supporting this issue – where addition of cement should be assumed first – we present Table 3, where some statistics is presented. It allows to evaluate what range of compressing strength can be expected if a certain percentage of cement is added.

Comment 3: How do you account for ramming energy? I would think that this is something that would be highly variable across different job sites. Is there any systematic study that has been done that would allow you to incorporate this also in the ANN?

Response: The dependence of the density of compacted mixture on energy used for compacting is obvious. The more energy is spent, the mixture density is closer to the maximum one (different for different aggregate content in a sample). We decided to use the density, as a representation of energy used for compaction. As found during literature review, there are variety of methods for compacting CSRE, so we have concentrated on the effect of the compaction i.e. density of the mixture.

Comment 3: The equations are shifted across two or more lines and it makes it difficult to read, please fix this. 

Response: The editing problem with formulas presented is fixed.

Reviewer 3 Report

The authors have used ANN in designing the composition of cement stabilized rammed earth. The ANN has already been used in many applications successfully. However, the major drawback is that the results obtained by individual authors may vary significantly. It is because different authors use different input and output parameters. Anyway, this is a good paper and it is well written. Few things as highlighted below need to consider revising the paper.

In introduction authors can briefly discuss about the application of ANNs in other fields. In this regard, some references are also recommended to add in the list. Authors may consider the following paper where ANN was used successfully to predict the concrete strength. A novel approach in modelling of concrete made with recycled aggregates, Measurement 2018, 115, 64-72.

How was the selection done for the training algorithm and the number of hidden neurons in ANN?

Is the soil compaction factor considered as one of the input for ANN? I don’t find it in fig 4.

Does this model valid for any grain size of the soil? The authors could recommend some the parameters for future research which may improve the accuracy of ANN.

Author Response

The authors are grateful to the Reviewer for appreciation of their effort, as well as, for valuable comments and suggestions.

Comment 1:  The authors have used ANN in designing the composition of cement stabilized rammed earth. The ANN has already been used in many applications successfully. However, the major drawback is that the results obtained by individual authors may vary significantly. It is because different authors use different input and output parameters. Anyway, this is a good paper and it is well written.

Response: Authors are grateful to the Reviewer for his/her positive and encouraging comment.

Comment 2:  In introduction authors can briefly discuss about the application of ANNs in other fields. In this regard, some references are also recommended to add in the list. Authors may consider the following paper where ANN was used successfully to predict the concrete strength. A novel approach in modelling of concrete made with recycled aggregates, Measurement 2018, 115, 64-72.

Response: The introduction is rewritten. The suggested reference is utilized (line 192).

Comment 3:  How was the selection done for the training algorithm and the number of hidden neurons in ANN?

Response: Based on features provided by Statistica developers, the training algorithm was chosen automaticly during the searching the ANN providing the lowest MSE. The authors limited the range of the number of neurons in the hidden layer to 3 – 30. From the possible set of activation functions (in Statistica software) only sinus was excluded. The software provide automatic searches for the parameters of ANN providing the lowest error, so it was used.

Comment 4:  Is the soil compaction factor considered as one of the input for ANN? I don’t find it in fig 4.

Response: The soil compaction factor is represented as one of inputs indirectly i.e. by a density. That is intuitive positive correlation: the higher the compaction factor, the higher the density is (up to the limit for the certain set of aggregates).

Comment 5:  Does this model valid for any grain size of the soil? The authors could recommend some the parameters for future research which may improve the accuracy of ANN.  

Response: Having the information the multi-layer perceptron (MLP) ANN predict with the higher level of accuracy than ANNs based on radial functions (RBF) It was decided to apply MLP type of ANN in hope that the model will be suitable for the other compositions of aggregates than tested by the authors. Nevertheless, we can recommend the method only for aggregate composition not exceeding much the band presented in Figure 6. As we stated at the end of conclusion the method proposed in the paper will be tested based on random soil samples obtained from different regions of our country. That will allow evaluating how far from the band (Figure 6) the new set of aggregates can be. The accuracy of predictions (based on validating set of data) is lower than preciseness of metering the water addition in real circumstances. So, we do not plan to work on increasing accuracy of ANN, but our plan is to confirm the method on random aggregate compositions. Then, the more universal

Reviewer 4 Report

See the attached document, which is the review for the Authors.

Author Response

The authors are grateful to the Reviewer for appreciation of their effort, as well as, for valuable comments and suggestions.

Comment 1:  The abstract seems to be the introduction. Moreover, it does not accurately describe the contents and does not include all of the main findings of the study. In the abstract of the revised version resubmitted, the following must be clear: the purpose of the work, the scope of the effort, the procedures used to execute the work, and the major findings.

Response: Abstract is rewritten.

Comment 2:  The introduction of the submitted manuscript does not provide sufficient background information for readers not in the immediate field to understand the problem, study, and results.

Response: The introduction is substantial rewritten. Areas of occurrence of numerous buildings made of earth are indicated. Historical achievements in applying a local construction materials are described in introduction to present the background for cement stabilized rammed earth technology. The description of the technology has been developed.

Comment 3:  Figure 1 must be better explained, including the role played by the strips shown (i.e., what the strips of the diagram represent).

Response: Figure 1 (now Figure 4) is re-drawn to make it easier understood.

Comment 4:  At page 4, row 78 (first row of the page), replace “constr4uction” with “construction”.

Response: The mistake in the word “construction” is removed.

Comment 5: The Authors must explain better the meaning of “- 0,289” and “- 0,622” in table 2, including the minus and the comma.

Response: The meaning of the values are explained in the text.

Comment 6:  The introduction of the revised version resubmitted has to provide a good, generalized background of the topic that quickly gives the reader an appreciation of the wide range of applications for this technology. However, to make the introduction more substantial, the Authors may wish to provide several references to substantiate the claim made, to make the motivation clearer, and to differentiate the paper some more from other applied papers. Cement stabilized rammed earth was used in the past too, while the Authors did not mention it. In particular, it was used in many Italian construction (e.g., in Venice, to build inner walls). Ultimately, the submitted manuscript lacks a thorough literature review. The introduction is one of the more difficult portions of a manuscript to write but also one of the most important too. Past studies are used to set the stage and provide the reader with information regarding the necessity of the represented project. Thus, the introduction must be rewritten. A competent introduction should include at least four key concepts:

1) significance of the topic,

2) the information gap in the available literature associated with the topic,

3) a literature review in support of the key questions,

4) subsequently developed purposes/objectives and hypotheses. In this case, the literature review must consider the uses of the proposed technology that were done in the past.

Response: Above mentioned remarks were carefully addressed.

Comment 7:  The papers that I suggest citing to improve the review of the state-of-the-art are the following two:

P. Foraboschi. Specific structural mechanics that underpinned the construction of Venice and dictated Venetian architecture. Engineering Failure Analysis, 2017; 78(August): 169-195.

P. Foraboschi. The central role played by structural design in enabling the construction of buildings that advanced and revolutionized architecture. Construction and Building Materials, 2016; 114(July): 956-976.

Those papers provide examples of constructions made of cement stabilized rammed earth that belongs to the history of architecture, including the proportions of soil, cement, and water that were used in the past.

Response: Suggested references are utilized.

Comment 8:  In the revised version resubmitted, the methodology must be sufficiently well explained that someone else knowledgeable about the field could repeat the study.

Response: To meet this requirements the paper is restructured and supplemented.

Comment 9:  Part of the Gaussian curve (which is also represented in Fig. 3) is negative. The Authors must explain that aspect.

Response: The Gaussian curve has been removed from Figure 3 (now Figure 7). It was drawn automatically by Statistica software (for the software compressive strength is a number only, can be negative). It was not an intention of the authors to fit the Gaussian curve to the results achieved. The aim of presenting this figure (as it is in a present form) is to make a reader familiar with the ranges of compressive strengths achieved during 373 samples tests.

Comment 10:  The format of Fig. 4 must be improved (the labels “yes” “no”, and “step” are written in bolt, capital, and large size characters, which is too much. 

Response: Figure 4 (now Figure 8) is redrawn.

Comment 11:  At page 10, row 227, the sentence “wy the value of activation function i.e. output from the neuron” is not clear and has no reference in the formula.

Response: Symbol “wy” is explained in the description to present Figure 9. (It was described under formula (2), where it is used too).

Comment 12:  The section devoted to the conclusion seems to be an abstract. In the revised version resubmitted, the Authors must provide sound and justified conclusions. Thus, I ask the authors to write a set of conclusions, or a summary and conclusion, in which the significant implications of the information presented in the body of the manuscript are reviewed.
When rewriting the conclusions, consider that the section is not just a restatement of your results, rather is comprised of some final, summative statements that reflect the flow and outcomes of the entire paper. Do not include speculative statements or additional material; however, based upon your findings a statement about potential changes in construction world or future research opportunities can be provided here.

Response: The abstract and conclusions are re-written and supplemented. Future researches are described in conclusions.

Comment 13: In the submitted manuscript, the study’s statement of purpose is not adequately illustrated. The revised version resubmitted must provide the study’s statement of purpose, including the role that cement stabilized rammed earth can play in modern construction and the reasons for performing this study.

Response: The role of cement stabilized rammed earth is now clearly illustrated in introduction and the purpose is clearly illustrated in abstract now.

Round  2

Reviewer 1 Report

Although part of my suggestions have not been applied in the revised verstion, I appreciate the authors great effort to follow them, considering at the same time the opinion of other reviewers and complying with the structure defined by the journal.

After reading the reviewed manuscript, I believe that it has been improved from the previous version and it may be accepted for publication in the present form.

Reviewer 4 Report

The Authors have performed a good job. They considered how I had commented their article and suitably addressed all my comments. Not only did they carefully and correctly addressed the issues raised in my review, but also they blended my suggestions or criticisms and further personal developments or contributions, which have further enriched the article.

Now, the article adds to the subject and the presentation saves the readers’ effort to understand the message that the article aims at conveying.

Thus, I recommend that the revised version of the article that has been resubmitted is accepted and published in the present form.